# Observational cohort study of the effect of a single lubricant exposure during transvaginal ultrasound on cell-shedding from the vaginal epithelium

D. Elizabeth O'Hanlon[1]☯, Sarah E. Brown[1,2]☯, Xin He[3], Christina A. Stennett[1,2], Sarah J. Robbins[1,2], Elizabeth D. Johnston[4], Amelia M. Wnorowski[5], Katrina Mark[6], Jacques Ravel[1,7], Richard A. Cone[8], Rebecca M. Brotman[1,2]*

1 Institute for Genome Sciences, University of Maryland School of Medicine, Baltimore, MD, United States of America, 2 Department of Epidemiology and Public Health, University of Maryland School of Medicine, Baltimore, MD, United States of America, 3 Department of Epidemiology and Biostatistics, University of Maryland, College Park, MD, United States of America, 4 Faculty Physicians Inc., University of Maryland, Baltimore, MD, United States of America, 5 Department of Diagnostic Radiology and Nuclear Medicine, University of Maryland School of Medicine, Baltimore, MD, United States of America, 6 Department of Obstetrics, Gynecology and Reproductive Sciences, University of Maryland School of Medicine, Baltimore, MD, United States of America, 7 Department of Microbiology and Immunology, University of Maryland School of Medicine, Baltimore, MD, United States of America, 8 Department of Biophysics, Johns Hopkins University, Baltimore, MD, United States of America

☯ These authors contributed equally to this work.
* rbrotman@som.umaryland.edu

**Data Availability Statement:** The GALE data can be found at the National Center for Biotechnology Information (NCBI) Database of Genotypes and

## Abstract

The outer layers of the vaginal epithelium (VE) are important because they accumulate glycogen which, under optimal conditions, *Lactobacillus* spp. consume to grow and acidify the vaginal microenvironment with lactic acid. We hypothesized that exposure to lubricant, for example in the conduct of a transvaginal ultrasound (TVUS), may contribute to the shedding of mature epithelial cells, exposing immature cells. Cervicovaginal fluid (CVF) was sampled at four time points by menstrual cup (Softdisc™) from 50 women referred for TVUS, during which a controlled volume of lubricant was applied to the TVUS wand. Samples were collected (1) immediately before TVUS and (2) 6–12 hours, (3) within one week, and (4) two weeks after TVUS. Clinical vaginal lubricants are similar to commercial lubricants, and often have a high osmolality or pH, and contain bactericides such as methylparaben and propylparaben. The number and maturity of epithelial cells in each CVF sample were measured by quantitative and differential fluorimetry (maturity index, MI). Comparisons of cell-counts and maturity were made by paired Wilcoxon signed-rank tests. Among women with a high pre-TVUS MI (> 3), there was a decrease in median cell-count and mean MI in the sample collected 6–12 hours after TVUS ($p<0.001$, $n = 26$ and $p < 0.001$, $n = 26$, respectively). For these women, cell-count and MI remained lower in the sample collected within the subsequent week ($p<0.001$, $n = 29$ and $p<0.01$, $n = 29$, respectively), and MI remained lower in the sample collected within two weeks of TVUS ($p<0.01$, $n = 25$), compared to the pre-TVUS sample. Among participants with a low pre-TVUS MI (< 3), cell-count was higher in the sample collected within two weeks of TVUS compared to the pre-TVUS sample ($p = $

Phenotypes (dbGaP) under accession number phs002211.

**Funding:** This study was funded by the National Institute of Allergy and Infectious Diseases (NIAID) R01-AI119012 (RMB). The funders had no role in study design, data collection and analysis, decision to publish, or preparation of the manuscript.

**Competing interests:** I have read the journal's policy and the authors of this manuscript have the following competing interests: J.R. is co-founder of LUCA Biologics, a biotechnology company focusing on translating microbiome research into live biotherapeutics drugs for women's health. All other authors declare that they have no competing interests. This does not alter our adherence to PLOS ONE policies on sharing data and materials.

0.03, $n = 15$), but no significant changes in MI were observed. Results were similar when restricted to reproductive-age women. This preliminary data indicates hypertonic vaginal lubricants may increase vaginal epithelial cell shedding.

## Introduction

The vaginal epithelium (VE) is the site of immunological [1], microbial [2] and biophysical [3] defenses against reproductive tract infections. The VE consists of approximately 30 layers of cells [4]. Cell proliferation in the basal zone pushes parabasal cells outwards [**Fig 1A**]. In the intermediate zone, maturing cells become flattened, heavily glycogenated, and organized into continuous sheets connected by desmosomes. In the superficial zone, dying cells detach from underlying layers and are shed singly or in aggregates. Glycogenated superficial cells and free glycogen in vaginal fluid may promote colonization by vaginal *Lactobacillus* spp. [5, 6], which produce lactic acid and acidify the vagina (pH <4) [7]. Lactic acid is a broad-spectrum bactericide and viricide [8–11], and also has a significant anti-inflammatory effect on the VE [12, 13]. Detachment/shedding of the outer layer of mature cells in excess of proliferation/maturation rates will expose the underlying living cells [**Fig 1C**], and it has been hypothesized that this may influence susceptibility or resistence to reproductive tract infections [14, 15]. The spermicide N9 causes an increase in detachment/shedding of epithelial cells that is associated with increased risk of HIV infection [16].

Personal lubricants are used to enhance sexual enjoyment [17, 18] and increase acceptability of condom-use [19, 20]. Lubricants also reduce the discomfort of clinical speculum examinations [21, 22] and are often recommended for symptomatic relief of some vulvovaginal symptoms associated with the genitourinary syndrome of menopause [23, 24]. Some lubricants have shown toxicity to epithelial cells *in vitro* [25–27], in a three-dimensional vaginal epithelium tissue-model [27–29] and in mice [30]. In most cases, the level of toxicity correlates with the hyperosmolality of the lubricant [31, 32] (i.e., the higher concentration of solutes in the lubricant compared with the tissues to which it is applied), as well as the presence of bacteriostatic preservatives (e.g. chlorhexidine, methylparaben and propylparaben).

Evaluating the effects of lubricant use with an observational study design (as compared to randomized trials) is challenging, as women use a variety of commercial lubricants with different composition, at varying doses and duration. We sought to study women presenting for transvaginal ultrasound because it provided the opportunity to observationally study lubricant use incidental to transvaginal ultrasound (TVUS), during which each woman is exposed to the same amount and type of lubricant that is consistently applied, and for approximately the same length of time. We hypothesized that exposure *in vivo* to a hyperosmolal vaginal lubricant would transiently increase the shedding of mature cells. Here, we report a longitudinal observational study which assessed cell shedding from the VE following a single exposure to lubricant gel during a TVUS exam.

## Materials and methods

### Participants

Eighty-six participants were recruited between May 2017 and May 2019 to take part in the Gynecology and Lubricant Effects (GALE) Study, a 10-week observational cohort study in Baltimore, MD which sought to investigate changes in the vaginal microenvironment among

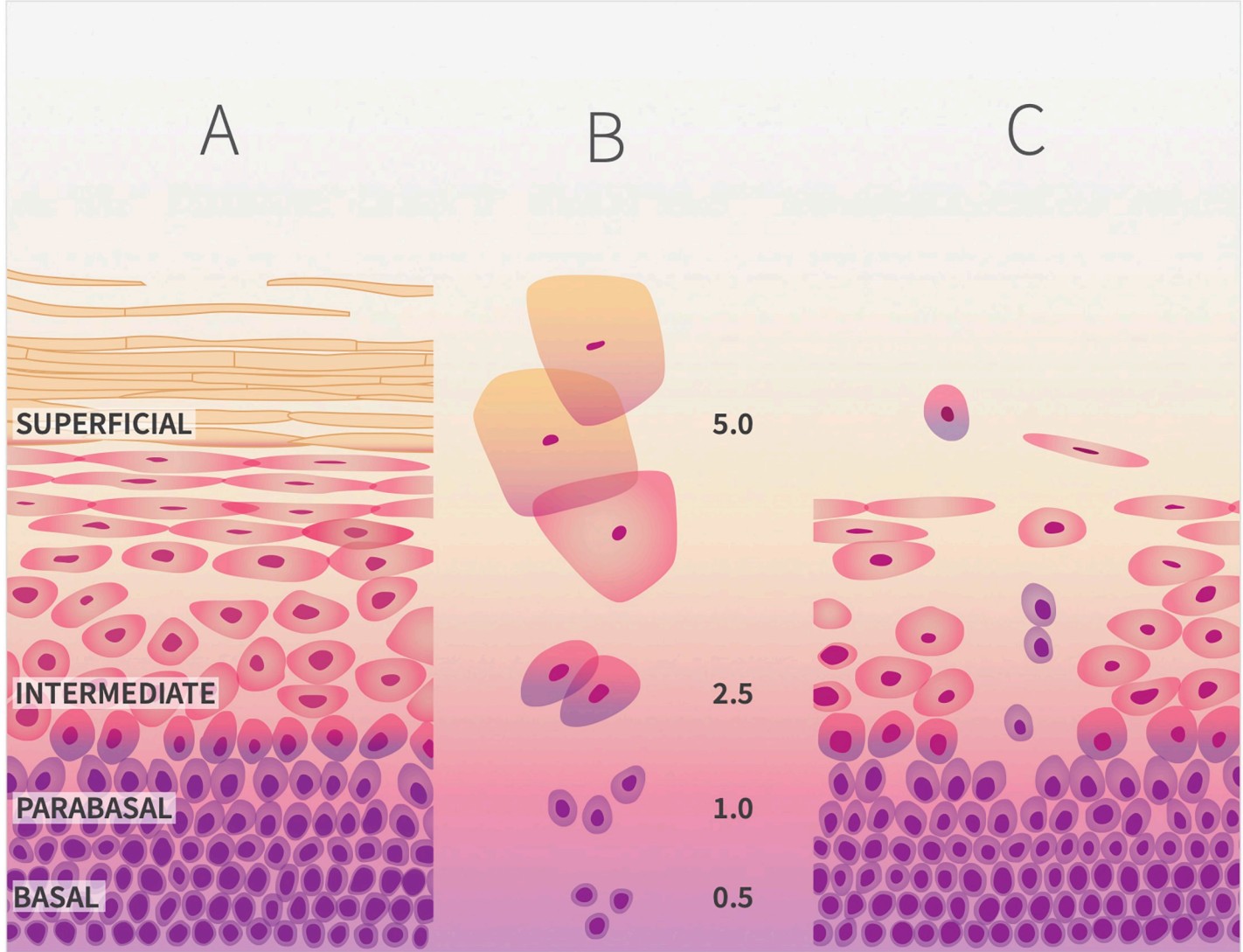

**Fig 1. A.** The vaginal epithelium undergoes morphological changes during maturation. As cells mature, they migrate from the basal layer, to the parabasal and intermediate layers to the superficial layer from which they are shed. **B.** Morphological changes of the vaginal epithelium during maturation include decrease in nuclear and increase in cytoplasmic sizes, which provides the basis of the Maturity Index value (values shown are representative of each layer). **C.** Shedding of immature cells indicates depletion of superficial cells, and loss of structural integrity of the vaginal epithelial barrier.

women exposed to a single use of approximately 3g of vaginal lubricant (E-Z™, Chester laboratory, 2,400 mOsm/kg and pH 5.5) during transvaginal ultrasound (TVUS). An a priori sample size calculation was not conducted for this secondary exploratory analysis. The GALE study was powered to detect an increase in molecular-BV [33] following TVUS.

Women over 18 years of age referred for TVUS at the Diagnostic Radiology and Nuclear Medicine Department at the University of Maryland Medical Center were eligible for the study. Reasons for referral included pelvic pain, pelvic mass and abnormal uterine bleeding [Table 1]. Exclusion criteria included cancer, diabetes, or other immunosuppressive conditions, diagnosis of laboratory-confirmed genital *Chlamydia trachomatis*, *Trichomonas vaginalis*, or *Neisseria gonorrhea* (BD MAX CT/GC/TV), syphilis (BD Macro-Vue), or HIV (Abbot ARCHITECT HIV Ag/Ab Combo), pregnancy, antibiotic or antifungal use in the month prior

**Table 1. Demographic and clinical description of study participants with a pre-TVUS epithelial cell maturity index (MI) less than or more than 3.**

| Characteristic | MI < 3 | | MI > 3 | | p-value |
|---|---|---|---|---|---|
| | N | Median | N | Median | |
| | (%) | (Q1-Q3) | (%) | (Q1-Q3) | |
| Age (years) | | 31 (27–51) | | 35 (31–40) | 0.82[c] |
| Menopausal status | | | | | 0.05[a] |
| Pre-menopausal | 12 (67%) | | 29 (91%) | | |
| Peri/post-menopausal | 6 (33%) | | 3 (9%) | | |
| Race | | | | | 0.69[a] |
| Black or African American | 12 (67%) | | 22 (69%) | | |
| White | 4 (22%) | | 6 (19%) | | |
| Asian | 2 (11%) | | 1 (3.1%) | | |
| Multiracial | 0 (0.0%) | | 2 (6.3%) | | |
| Other | 0 (0.0%) | | 1 (3.1%) | | |
| Current use of hormonal contraception | 5 (29%) | | 13 (42%) | | 0.65[b] |
| *Type of hormonal contraception* | | | | | |
| Oral contraceptive | 2 (11%) | | 4 (13%) | | |
| IUD (progestin) | 2 (11%) | | 4 (13%) | | |
| Implant | 1 (6%) | | 5 (16%) | | |
| Menses reported on day of TVUS | 3 (17%) | | 4 (13%) | | 0.70[a] |
| TVUS Indications | | | | | |
| Assessment of pelvic mass | 6 (33%) | | 11 (34%) | | 0.94[b] |
| Localization of intrauterine device | 2 (11%) | | 3 (9.4%) | | 0.99[a] |
| Abnormal uterine bleeding | 1 (5.6%) | | 10 (31%) | | 0.07[a] |
| Screening for malignancy | 0 (0.0%) | | 1 (3.1%) | | 0.99[a] |
| Pelvic pain | 10 (56%) | | 11 (34%) | | 0.14[b] |
| Other | 3 (17%) | | 1 (3.1%) | | 0.13[a] |
| TVUS Findings | | | | | |
| Adenomyosis | 0 (0.0%) | | 3 (9.4%) | | 0.54[a] |
| Fibroids | 4 (22%) | | 9 (28%) | | 0.75[a] |
| Cysts | 1 (5.6%) | | 5 (16%) | | 0.40[a] |
| Other | 4 (22%) | | 8 (25%) | | 0.99[a] |
| No significant findings | 11 (61%) | | 8 (25%) | | 0.01[b] |

The median weight of CVF recovered by clinician collection at W1-post-TVUS was 86 mg (IQR 28–172 mg); the median weight recovered by participant self-collection at W2-post-TVUS was 62 mg (IQR 9–176 mg). There was no significant difference in weight comparing paired clinician-collected and participant-collected samples at W1-post-TVUS and W2-post-TVUS, respectively, p = 0.81.

Differences across groups assessed by Fisher's exact test (a), chi-square test (b), and Wilcoxon rank sum test (c).

to TVUS, and self-reported lubricant use in the week prior to TVUS. Participants self-collected vaginal samples during the week prior to TVUS and for nine weeks following TVUS, and attended three clinical visits at enrollment, within one week of TVUS, and at the completion of the study. No lubricant was used during the three clinical visits, and sample collection occurred prior to the pelvic exam. Participants also completed daily sexual health and behavior diaries and met with study coordinators weekly to drop off samples, which they stored in their home freezers and transported to appointments in a cooler with an ice-pack. Participants gave written informed consent under a protocol approved by the Human Research Protection Office at the University of Maryland Baltimore.

## Sample collection and processing

Participants' cervicovaginal fluid (CVF) was sampled (1) immediately before TVUS (pre-TVUS), (2) at bedtime (approximately 6–12 hours after TVUS) on the day of the ultrasound, (post-TVUS), (3) at a follow-up visit within one week of TVUS (W1-post-TVUS) and (4) within two weeks of TVUS (W2-post-TVUS). CVF was sampled using a disposable menstrual device, the Softdisc™ (The Flex Company, Venice CA). Samples were self-collected by participants except for the W1-post-TVUS sample, which was collected by the clinician at the post-TVUS follow-up visit. As previously described [34], this sampling method permits collection and recovery of relatively large volumes of undiluted CVF. At each sampling, a Softdisc™ was inserted into the vagina and immediately removed, then placed in a 50 mL plastic conical tube. Tubes were immediately frozen at -20˚C in the participants' home freezer and transported frozen to the clinical visits, where they were stored at -20˚C until processing (typically less than three weeks). Previous control experiments suggested that freezing undiluted vaginal fluid samples did not affect cell count or maturity index scores compared to fresh condition (data not shown).

At processing, each tube was thawed at room temperature for ten minutes, then centrifuged at 1250 *rpm* for three minutes to recover the collected CVF from the Softdisc™. The Softdisc™ was discarded. Approximately 50 µL of the collected CVF was transferred, using a wide bore pipette tip, to a tared 1.5 mL Flex-Tube® (Eppendorf North America, Hauppauge NY). Each tube was reweighed and the exact mass of the CVF sample was calculated. Each sample was diluted with 1 mL of PBS and centrifuged at 10,000 *rpm* for ten minutes. The supernatant was drawn off and each pellet was resuspended in 1 mL trypsin solution (0.5% v/v trypsin-EDTA in PBS). Tubes were incubated at 37˚C for 40 minutes and vortexed for a few seconds every ten minutes. After incubation each tube was vortexed again, and 3 µL of each trypsinized suspension was added to 3 µL of trypan blue solution (0.4% w/v trypan blue in PBS) in a well of a 12-well microscope slide (Tekdon Inc., Myakka City FL). Complete trypsinization was confirmed by examination at 100x total magnification (i.e., the preparation was a single-cell suspension without cell clumps). The tubes were then centrifuged at 10,000 *rpm* for five minutes. The supernatant was drawn off as completely as possible without disturbing the pellet. Using a 1 µL to 1 mg equivalence, and the exact mass of each CVF sample processed, ten volumes of PBS were added to the pellet and the pellet thoroughly resuspended by gently pipetting. Of the approximately 500 µL of trypsinized cells in PBS produced, a 100 µL aliquot was stored at -20˚C for later measurement of cell maturity (typically less than three months).

## Cell count measurements

The number of cells was measured using bulk fluorimetry and fluorescence microscopy. A 100 µL aliquot of the freshly trypsinized cells in PBS was transferred to a fresh 1.5 mL Flex-Tube®; 100 µL of SYTO™ 13 (Molecular Probes, Eugene OR) staining solution (0.5% v/v SYTO™ 13 in TAE) was added to each tube. The tubes were incubated for 40 minutes, at room temperature protected from light. An 80 µL aliquot of stained cell suspension was loaded into each of two replicate wells of a 96-well spectrophotometry plate. Fluorescence was measured at 488 nm$_{ex}$510nm$_{em}$. The measured fluorescence emissions of the two replicates were averaged. To convert fluorescence emission values to cell counts, the remaining 40 µL of stained cells were diluted with TAE to bring the measured fluorescence to approximately 100 (at which the concentration of cells becomes almost countable by microscopy) and the factor of dilution was noted. A further serial three-fold dilution with TAE was made, to give a total of six descending concentrations. A 10 µL aliquot of each concentration was spotted into two replicate wells of a 12-well microscope slide; the slides were partially air-dried, at room temperature and

protected from light. Each well was surveyed at 100x total magnification, and individual cells were counted if the well's cell counts were not too numerous (< approximately 250 cells per well). Every sample yielded at least two wells with countable numbers of cells: the cell count of each well was corrected for the three-fold dilution and the counts averaged. This value was further corrected for the variable dilution factor (necessary to reach 488 nm$_{ex}$510nm$_{em}$ of approximately 100), to give the estimated cell count in the samples measured by bulk fluorimetry and finally corrected ten-fold to give the cell count in 10 μL native CVF sample.

## Cell maturity measurements

As VE cells mature, the ratio of cytoplasm to nucleus increases [35] [**Fig 1B**]. Cell maturity in CVF was measured using differential fluorescence staining of cytoplasm and nuclei [36]. Frozen 100 μL aliquots of trypsinized cells in PBS were thawed on ice and 500 μL of chilled standard saline containing 20% v/v ethanol and 0.025% w/v thymol was added prior to incubation at 4˚C for approximately 24 hours, after which the liquid was removed. One mL of acridine orange solution (0.010% w/v in water) was added. The mixture was incubated for ten minutes at room temperature and transferred to a 15 mL plastic conical tube containing 12 mL of 5M salt solution. After centrifugation at 5,000 *rpm* for five minutes, the liquid was poured off. The cell pellet was resuspended in 400 μL of standard saline. A 180 μL aliquot of this suspension was loaded into two replicate wells of a 96-well spectrophotometry plate. Fluorescence was measured at 488nm$_{ex}$650nm$_{em}$ (cytoplasm) and at 488nm$_{ex}$550nm$_{em}$ (nuclei). The ratios of emission$_{650nm}$ to emission$_{550nm}$ for the two replicates were averaged and reported as Maturity Index (MI).

## Statistical analyses

The exposure was lubricant use during TVUS. Cell-count and MI of each pre-TVUS sample were compared, pairwise, to each participant's cell-count and MI of her corresponding post-TVUS, within one week (W1-post-TVUS) and within two weeks (W2-post-TVUS) samples. Comparisons of cell-count and MI are presented as median difference (interquartile range Q1-Q3) and *p* value from nonparametric Wilcoxon signed rank tests. Not all women provided samples with a useable volume of CVF at all four time points (22% at post-TVUS, 16% at W1-post-TVUS, and 18% at W2-post-TVUS did not have usable volume), therefore *n* is reported separately for each comparison.

Analysis was conducted in aggregate for all participants, as well as stratified by baseline MI, as baseline MI may restrict the scope of further MI changes. Baseline MI was categorized as low (MI < 3) or high (MI > 3); MI = 3 approximates an equal prevalence of mature and immature cells in a sample. No samples had a MI = 3. Baseline demographic and clinical characteristics were compared between women with low and high pre-TVUS MI using chi-square or Fisher's exact tests for categorical variables and Wilcoxon rank sum tests for continuous variables. At W2-post-TVUS, cell-count and MI were also compared between women with low versus high pre-TVUS MI using Wilcoxon rank sum tests. A sensitivity analysis was performed by repeating the analysis only with samples from reproductive-age participants. For this analysis, pre-, peri- and post-menopause status was defined by a combination of variables including age, STRAW +10 menstrual cycle criteria [37], and use of hormonal contraception.

## Results

### Study participant characteristics

Eighty-six women were enrolled in the GALE study, of whom 56 provided samples at all four time points, and 50 contributed a useable sample to determine baseline MI and cell-count for

pairwise comparisons. The most common indication for TVUS referral was pelvic pain, followed by assessment of pelvic mass. The most common diagnosis from TVUS was fibroids (13/50, 27%), but a plurality of participants (19/50, 36%) had no significant finding from TVUS. The median number of days between collection of the pre-TVUS sample and the W1-post-TVUS and W2-post-TVUS samples were 4 days (IQR: 3–5 days) and 7 days (IQR: 6–9 days), respectively.

There were a few differences in baseline factors between women with high MI versus low MI at the pre-TVUS sampling (**Table 1**). The median age was not statistically different between the high pre-TVUS MI (31 years) vs low pre-TVUS MI (35 years), although women in the low pre-TVUS MI group were more likely to be peri/post-menopausal. Abnormal uterine bleeding as the indication for referral for TVUS was more common in women with high pre-TVUS MI, although women with a low MI were significantly more likely to have no findings on TVUS. There was no association between hormonal contraception use or menses reported on TVUS day and pre-TVUS MI. Both groups reported similar patterns of sexual activity and menses based on analysis of behavioral diaries on sampling days; however, receptive oral sex was more commonly reported by women with low pre-TVUS MI than by women with high pre-TVUS MI (12% versus 3% of sampling days, respectively, p = 0.03).

### Within-woman comparisons

**Cell counts.** The pre-TVUS median cell count of samples was 17,421/10 μL (IQR 3,616/10 μL to 24,225/10 μL), and the median pre-TVUS MI was 3.29 (IQR 2.01 to 4.11). We found that post-TVUS samples taken 6–12 hours after TVUS contained significantly fewer VE cells than pre-TVUS samples, with a median change in cell count of -6,693/10 μL (IQR -19,768/10 μL to 63/10 μL), $p = 0.02$, $n = 39$ [**Fig 2A**]. There was no significant difference in cell count between pre-TVUS samples and W1-post-TVUS samples taken within one week of TVUS, which had a median change in cell count of -1,968/10 μL (IQR -13,112/10 μL to 3,171/10 μL), $p = 0.1$, $n = 42$. Similarly, there was not a significant difference in cell count between pre-TVUS samples and W2-post-TVUS samples taken within two weeks of TVUS, which had a median change in cell count of 879/10 μL (IQR -5,383/10 μL to 11,273/10 μL), $p = 0.2$, $n = 40$.

**Maturity Index (MI).** Considering all participants together, the MI of post-TVUS samples was significantly lower than that of pre-TVUS samples, with a median change of -0.93 (IQR -2.15 to -0.02), $p < 0.01$, $n = 39$ [**Fig 2B**]. The difference in MI between pre-TVUS samples and W1-post-TVUS samples was significant, with a median change of -1.01 (IQR -2.29 to 0.45), $p = 0.03$, $n = 42$. The difference in MI between pre-TVUS samples and W2-post-TVUS samples was not significant, with a median change of -0.54 (IQR -1.76 to 0.58), $p = 0.1$, $n = 40$.

Because the state of the epithelium before exposure to lubricant might affect findings, we stratified analysis into two groups: those whose pre-TVUS MI was less than 3 ($n = 18$), indicating a paucity of mature superficial cells and predominance of immature intermediate and parabasal cells, and those whose pre-TVUS MI was greater than 3 ($n = 32$), indicating predominance of mature superficial cells.

### Analysis of women with high pre-TVUS Maturity Index

**Cell counts.** Among participants with high pre-TVUS MI ($> 3$), post-TVUS samples contained significantly fewer VE cells than pre-TVUS samples, with a median change in cell count of -16,690/10 μL (IQR -20,130/10 μL to -4,997/10 μL), $p < 0.001$, $n = 26$ [**Fig 2C**]. Among these participants, W1-post-TVUS samples also contained significantly fewer VE cells than pre-TVUS samples, with a median change in cell count of -6,491/10 μL (IQR -14,439/10 μL to 1,268/10 μL), $p = 0.01$, $n = 29$. However, there was no significant difference between the cell

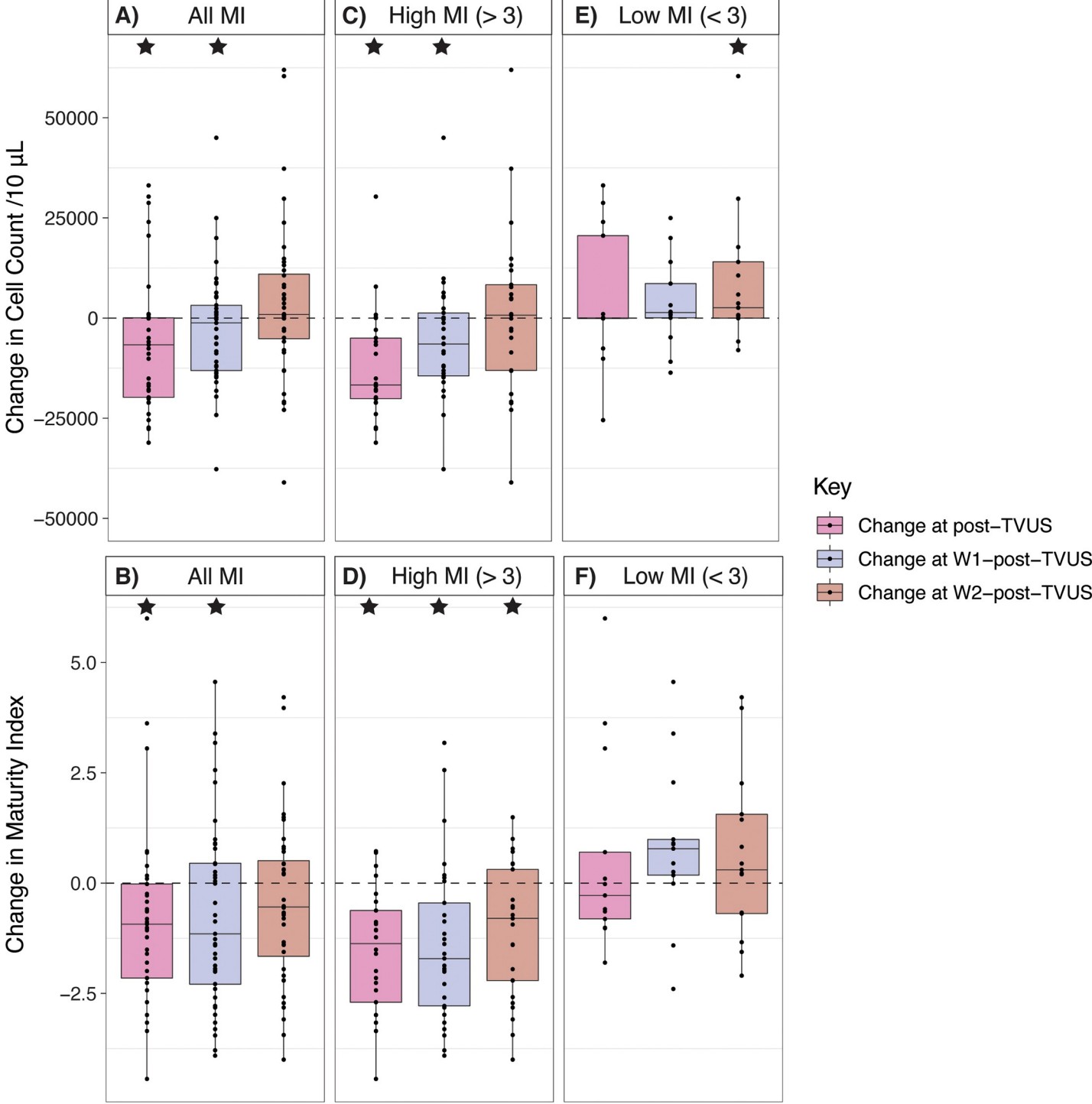

**Fig 2. Dashed lines represent change = 0.** Stars represent significant pairwise differences from pre-TVUS samples. **A.** For all participants, the change in VE cell count per 10 μL CVF in samples taken approximately 12 hours later (post-TVUS), within one week (W1-post-TVUS), and within two weeks (W2-post-TVUS), compared to immediately before TVUS. **B.** For all participants, the change in maturity index (MI) of samples taken approximately 12 hours later (post-TVUS), within one week (W1-post-TVUS), and within two weeks (W2-post-TVUS), compared to immediately before TVUS. **C.** For only participants with a pre-TVUS MI > 3, the change in VE cell count per 10 μL CVF in samples approximately 12 hours later (post-TVUS), within one week (W1-post-TVUS), and within two weeks (W2-post-TVUS), compared to immediately before TVUS. **D.** For only participants with a pre-TVUS MI > 3, the change in maturity index (MI) of samples taken approximately 12 hours later (post-TVUS), within one week (W1-post-TVUS), and within two weeks (W2-post-TVUS), compared to immediately before TVUS. **E.** For only participants with a pre-TVUS MI < 3, the change in VE cell count per 10 μL CVF in samples approximately 12 hours later (post-TVUS), within one week (W1-post-TVUS), and within two weeks (W2-post-TVUS), compared to immediately before TVUS. **F.** For only participants with a pre-TVUS MI < 3, the change in maturity index (MI) of samples taken approximately 12 hours later (post-TVUS), within one week (W1-post-TVUS), and within two weeks (W2-post-TVUS), compared to immediately before TVUS.

count of pre-TVUS samples and W2-post-TVUS samples, with a median change of 731/10 μL (IQR -13,058/10 μL to 8,317/10 μL), $p$ = 0.9, $n$ = 25.

**Maturity Index (MI).** Among participants with high pre-TVUS MIs ($> 3$), the MI of post-TVUS samples was significantly lower than that of pre-TVUS samples, with a median change of -1.37 (IQR -2.70 to -0.62), $p < 0.001$, $n$ = 26 [**Fig 2D**]. The MI of W1-post-TVUS samples was significantly lower than that of pre-TVUS samples, with a median change of -1.71 (IQR -2.78 to -0.45), $p < 0.01$, $n$ = 29; the MI of W2-post-TVUS samples was also significantly lower than that of pre-TVUS samples, with a median change of -0.80 (IQR -2.21 to 0.31), $p < 0.01$, $n$ = 25.

## Analysis of women with low pre-TVUS Maturity Index

**Cell counts.** Among participants with low pre-TVUS MIs ($< 3$), there was no significant difference in cell count between pre-TVUS and post-TVUS samples, with a median change of 25/10 μL (IQR -85/10 μL to 20,560/10 μL), $p$ = 0.5, $n$ = 13 [**Fig 2E**]. There was no significant difference in cell count between pre-TVUS sample and W1-post-TVUS samples, with a median change of 1,379/10 μL (IQR 56/10 μL to 8,608), $p$ = 0.2, $n$ = 13. However, there was a significant increase in cell count from pre-TVUS samples to W2-post-TVUS samples, with a median change of 2,586/10 μL (IQR -5/10 μL to 14,031/10 μL), $p$ = 0.03, $n$ = 15.

**Maturity Index (MI).** Among participants with low pre-TVUS MI ($< 3$), there was no significant difference in MI between pre-TVUS sample and post-TVUS samples, with a median change of -0.28 (IQR -0.81 to 0.70), $p$ = 0.9, $n$ = 13 [**Fig 2F**]. Similarly, there was no significant difference in MI between pre-TVUS samples and W1-post-TVUS samples, with a median change of 0.78 (IQR 0.18 to 0.99), $p$ = 0.09, $n$ = 13; or between pre-TVUS samples and W2-post-TVUS samples, with a median change of 0.30 (IQR -0.69 to 1.56), $p$ = 0.3, $n$ = 15.

## Analysis of reproductive-age women

**Cell counts.** Among 12 reproductive-age participants with low pre-TVUS MI ($< 3$), there was no significant difference in cell count between pre-TVUS samples and either the post-TVUS, W1-post-TVUS, or W2-post-TVUS samples [**S1 Table**]. Among 30 reproductive-age participants with high pre-TVUS MI ($> 3$), cell counts were significantly decreased in the post-TVUS and W1-post-TVUS samples, with a median decrease in cell count of 16,452/10 μL and 6444/10 μL, respectively.

**Maturity Index (MI).** Among 12 reproductive-age participants with low pre-TVUS MI ($< 3$), there was no significant difference in MI between pre-TVUS samples and either the post-TVUS, W1-post-TVUS, or W2-post-TVUS samples. Among 30 reproductive-age participants with high pre-TVUS MI ($> 3$), MI was significantly decreased in the post-TVUS, W1-post-TVUS, and W2-post-TVUS samples, with a median decrease of 1.51, 1.79, and 0.72, respectively.

## Comparison of women with high and low pre-TVUS Maturity Index at W2-post-TVUS

To assess whether there was any association between the initial condition of the epithelium and the condition of the epithelium two weeks following lubricant exposure, we also compared W2-post-TVUS samples from women with high pre-TVUS MI to samples from women with low pre-TVUS MI.

**Cell count.** The median cell-count of W2-post-TVUS samples from women with low pre-TVUS MIs was not significantly different than that of women with pre-TVUS MIs $> 3$, $p$ = 0.2, $n$ = 40 [**Fig 3A**].

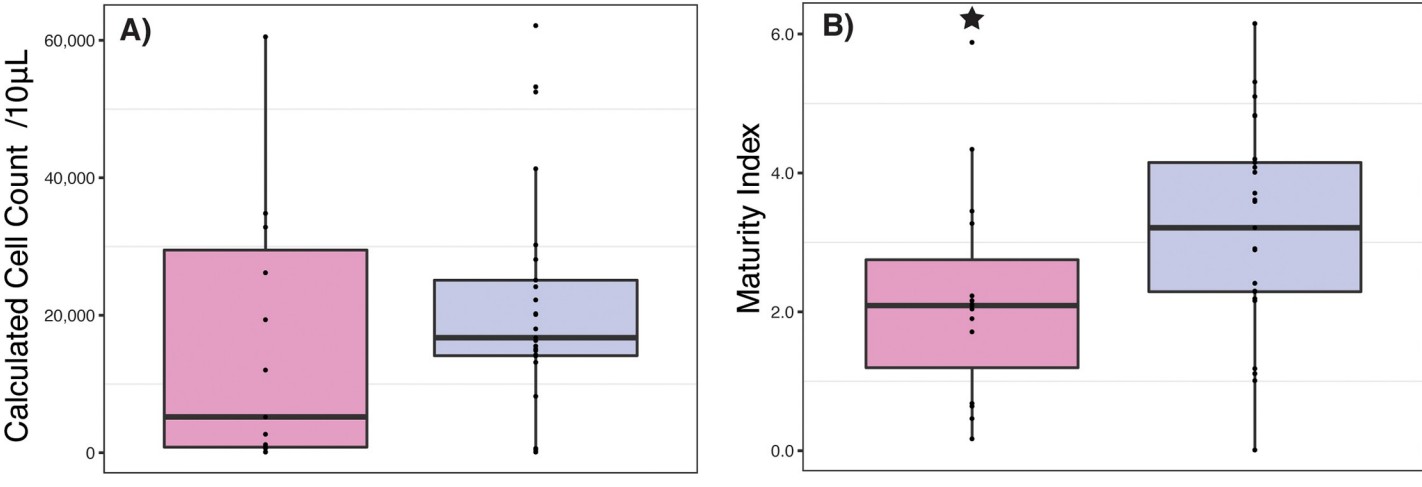

**Fig 3. Stars represent significant difference between low MI and high MI samples. A.** Cell-count per 10 μL CVF in samples taken within two weeks of TVUS (W2-post-TVUS) with lubricant exposure, comparing women with a pre-TVUS maturity index < 3 and women with a pre-TVUS maturity index > 3. **B.** Maturity index of samples taken within two weeks of TVUS (W2-post-TVUS) with lubricant exposure, comparing women with a pre-TVUS maturity index < 3 and women with a pre-TVUS maturity index > 3.

**Maturity Index (MI).** The MIs of W2-post-TVUS samples from women with low pre-TVUS MIs were significantly lower than that from women with high pre-TVUS MIs, with a median W2-post-TVUS MI of 2.09 (IQR 0.68 to 3.27) in women with low pre-TVUS MI and median W2-post-TVUS MI of 3.21 (IQR 2.29 to 4.15) in women with high pre-TVUS MIs, $p = 0.02$, $n = 40$ [**Fig 3B**].

Plots of individual trajectories for MI and cell counts are included in **S1 Fig**.

## Discussion

We hypothesized that a single vaginal lubricant exposure, as part of a TVUS protocol, would promote shedding of VE cells, resulting in loss of fully mature upper layers of VE cells and increased exposure of immature intermediate and parabasal cells.

Our original expectation was that CVF samples taken approximately 12 hours after TVUS would contain more VE cells than samples taken immediately before TVUS, indicating increased detachment/shedding. In fact, participant's post-TVUS samples contained significantly *fewer* cells than their pre-TVUS samples, and post-TVUS samples also had a significantly lower MI than paired pre-TVUS samples. One possible explanation for these findings is that rapid detachment/shedding and subsequent clearance of VE cells may have occurred very soon after lubricant exposure, resulting in the observed decreases in cell maturity. This rapid loss hypothesis is supported by published *in vivo* research showing that rectal epithelial denudation caused by hyperosmolal lubricants increases within 1.5 hours of lubricant application [32]. Future studies could include additional sampling time-points immediately following exposure to lubricant to investigate the rapidity of the shedding response.

The osmolality of the lubricant used in this study (approximately 2,400 mOsm/kg [27]) is in the middle of the range for personal lubricants [32] and the length of exposure (all TVUS procedures were completed in approximately 30 minutes) is reasonably consistent with use of lubricants for sexual activities. The effects observed in this study are, therefore, likely to pertain

to non-clinical use of many other personal lubricants as well. Additionally, while it is not possible with this observational study design to separate the effects of lubricant from other possible factors, including effects of the TVUS probe or SoftDisc™ collection device, *in vitro* studies support that hyperosmolal lubricants alone are capable of reducing epithelial cell barrier integrity and causing tissue morphological damage [29, 38].

The epithelial cell changes observed in this study occurred almost entirely in participants whose pre-TVUS sample had high MI ($> 3$). It has been hypothesized that exfoliation of mature superficial cells is one mechanism by which the vaginal epithelium provides protection against infection [14]. However, shedding of immature cells reflects a depletion of superficial cells and a loss of vaginal epithelial barrier integrity. Following the single use of vaginal lubricant during TVUS, we observed the shedding of fewer, and more immature, VE cells in both the W1-post-TVUS and W2-post-TVUS samples, suggesting this proposed protective mechanism may be disrupted.

MI did not change significantly in samples from participants with a low pre-TVUS MI ($< 3$). VE cell-count was significantly increased in the W2-post-TVUS sample, but not in the post-TVUS or the W1-post-TVUS samples. In other words, in women whose VE is already depleted of mature cells, lubricant exposure does not appear to produce much discernable change in the amount or maturity of cells shed as measured by our methodology. Indeed, W2-post-TVUS samples from these women had a significantly lower MI than W2-post-TVUS samples from women with pre-TVUS MI $> 3$, suggesting that there may be a prior cause for low MI in these women that persisted. Although there was no association between hormonal contraception use or menstrual bleeding and pre-TVUS MI, the low pre-TVUS MI group did have a significantly higher percentage of post-menopausal women, consistent with the expected morphology of the post-menopausal vaginal epithelium [39]. However, when restricting the analysis to reproductive-age women, there were still no significant changes in MI or cell-count among women with a low pre-TVUS MI. Overall, this finding argues against the change in women with high pre-TVUS MI being due to regression to the mean, in which case one would expect to see an equally prevalent *increase* in MI among women with a low pre-TVUS MI.

A limitation of this study is that we were unable to assess the effect of any pre-study lubrication products. If pre-TVUS samples were affected by prior lubricant use, this could lead to an underestimation of the effect of lubricant used during TVUS on the vaginal epithelium. However, the risk of bias is minimized due to study exclusion criteria that prohibited the use of lubricant in the week prior to TVUS, and this non-use period was longer for most participants. Eighty-four percent reported they had not used a lubricant in the two months prior to enrollment. In addition, the use of a cohort of women seeking clinical care may affect the generalizability of these results to all women, although 37% of participants had no significant findings from TVUS. Fibroids were a common finding on TVUS; however, there are no data relating fibroids to the vaginal epithelium. The vaginal microbiota was not evaluated, and we have previously shown that asymptomatic bacterial vaginosis (BV) is associated with the shedding of fewer, lower MI epithelial cells [15]. While menses did not appear to significantly influence our analysis and there was no association between hormonal contraception and baseline MI, we were unable to model menstrual cycle phase or specific types of hormonal contraception due to sample size limitations. Patton *et al* found evidence for only small changes in the mean number of epithelial cell layers throughout the menstrual cycle (27.8 on days 1–5, 28.1 on days 7–12, and 26.0 on days 19–24) [4]. Future studies could explore the association between lubricant use and the vaginal epithelium controlling for a wide variety of baseline and time-varying factors. Lastly, we were unable to make comparisons with pre-TVUS MI and cell-count for 22%, 16%, and 18% of samples at the post-TVUS, W1-post-TVUS, and W2-post-TVUS

timepoints because the SoftDisc$^{TM}$ did not recover a large enough sample volume for experiments. It is possible that removal of VE cells by the SoftDisc at the pre-TVUS sample may have contributed to the higher proportion of unusable samples at the post-TVUS timepoint. 45% of samples without a usable volume occurred among women with a low pre-TVUS MI, and 55% occurred among women with a high pre-TVUS MI.

Strengths of this study include the longitudinal within-woman analysis, collection of confounding time-varying behavioral data that might affect results and uniform exposure of each woman to the same volume of lubricant product, used in the same way for approximately the same duration of exposure.

## Conclusions

These preliminary data raise potentially important questions about the effect of lubricant on the vaginal epithelium. If, as our findings suggest, even a single lubricant exposure during transvaginal ultrasound can cause significant rapid loss of mature VE cells and exposure of immature ones, then lubricant exposure may meaningfully decrease VE barrier integrity. To our knowledge, no direct measurements of the VE have been made before and after lubricant exposure, perhaps one can envision using optical coherence tomography (OCT) which can accurately measure the thickness of the VE *in vivo* without disrupting its integrity [40]. Future studies should explore whether changes in VE barrier integrity might be associated with an increased inflammation and susceptibility to reproductive tract infections. Certainly, our findings support the importance and urgency of finding less cytotoxic formulations for lubricants and other related feminine products.

## Supporting information

**S1 Table. Wilcoxon signed-rank test to evaluate changes in maturity index and cell count from pre-TVUS to post-TVUS, W1-post-TVUS, and W2-post-TVUS among reproductive age women, stratified by pre-TVUS maturity index.**
(DOCX)

**S1 Fig. A and B.** Plots of individual trajectories for cell-counts and MI, respectively, in women with a pre-TVUS maturity index > 3. **C and D.** Plots of individual trajectories for MI and cell counts, respectively, in women with a pre-TVUS maturity index < 3.
(TIF)

## Author Contributions

**Conceptualization:** D. Elizabeth O'Hanlon, Jacques Ravel, Richard A. Cone, Rebecca M. Brotman.

**Formal analysis:** Sarah E. Brown, Xin He.

**Funding acquisition:** Katrina Mark, Jacques Ravel, Richard A. Cone, Rebecca M. Brotman.

**Investigation:** D. Elizabeth O'Hanlon, Christina A. Stennett, Sarah J. Robbins, Elizabeth D. Johnston.

**Methodology:** D. Elizabeth O'Hanlon.

**Supervision:** Amelia M. Wnorowski, Katrina Mark, Rebecca M. Brotman.

**Writing – original draft:** D. Elizabeth O'Hanlon, Sarah E. Brown.

**Writing – review & editing:** Christina A. Stennett, Sarah J. Robbins, Elizabeth D. Johnston, Jacques Ravel, Richard A. Cone, Rebecca M. Brotman.

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
