## [Decision Letter · Decision Letter 0]

1 Oct 2020

PONE-D-20-18637

Observational cohort study of the effect of a single lubricant exposure on cell-shedding from the vaginal epithelium

PLOS ONE

Dear Dr. Brotman,

Thank you for submitting your manuscript to PLOS ONE, and thank you for your patience in what has been a long review process. After careful consideration, we feel that it has merit but does not fully meet PLOS ONE’s publication criteria as it currently stands. Therefore, we invite you to submit a revised version of the manuscript that addresses the points raised during the review process.

You will see that Reviewer #1 has raised major concerns around the potential effect of confounders on the validity of the conclusions drawn, in particular the ability without a control group to attribute changes to the lubricant use rather than prior application of the softcup or the ultrasound probe. They also felt that the data were not sufficient to conclude that lubricant use "may increase infection susceptibility", and raised concerns around the effect of hormonal milieu and the microbiome. Any resubmission will need to deal with these concerns around the validity of conclusions very carefully.

We look forward to receiving your revised manuscript.

Kind regards,

Rupert Kaul

Academic Editor

PLOS ONE

Journal Requirements:

2. Please refer to any sample size calculations performed prior to participant recruitment. If these were not performed please justify the reasons. Please refer to our statistical reporting guidelines for assistance (https://journals.plos.org/plosone/s/submission-guidelines.#loc-statistical-reporting).

"I have read the journal's policy and the authors of this manuscript have the following competing interests: J.R. is co-founder of LUCA Biologics, a biotechnology company focusing on translating microbiome research into live biotherapeutics drugs for women’s health. All other authors declare that they have no competing interests"

Reviewers' comments:

Reviewer's Responses to Questions

**Comments to the Author**

1. Is the manuscript technically sound, and do the data support the conclusions?

Reviewer #1: No

Reviewer #2: Yes

2. Has the statistical analysis been performed appropriately and rigorously? 

Reviewer #1: Yes

Reviewer #2: Yes

3. Have the authors made all data underlying the findings in their manuscript fully available?

Reviewer #1: Yes

Reviewer #2: Yes

4. Is the manuscript presented in an intelligible fashion and written in standard English?

Reviewer #1: Yes

Reviewer #2: Yes

5. Review Comments to the Author

Reviewer #1: Review for PLOS ONE (PONE-D-20-18637)

Title: Observational cohort study of the effect of a single lubricant exposure on cell-shedding from the vaginal epithelium

This study is a non-randomized, single group cohort study of non-pregnant women 18+ years undergoing transvaginal ultrasound for a variety of reasons and willing to collect vaginal fluid using a softcup immediately prior to and at times following the transvaginal ultrasound. The authors evaluated the collected vaginal fluid for number maturity index of epithelial cells. Overall, I have several major concerns about premise as outlined by the authors and the study design, which suffers from exposure confounders that impede interpretation of the results.

1. The opening premise is that “the other layers of the vaginal epithelium form a physical barrier of dead and dying cells that protect the inner maturing cell layers.” It is unclear what this premise is based on as there are no references provided to this specific point. This actually seems to be a hypothesis for the function of the outer, dead layers of the vaginal epithelium. Other hypotheses also exist, such as that these layers provide a food source for commensal bacteria and/or that they don’t have a specific function and rather simply are a byproduct of the epithelial cell maturation and sloughing process.

2. The authors hypothesize that a single vaginal exposure to commonly used lubricating jelly “accelerates” shedding of the vaginal epithelium. However, in order to test “acceleration” then there must be a comparison to a non-exposed group. The study was not designed this way, rather the women all had multiple variables that may have impacted the measured outcomes (epithelial cell number and maturation index): 1. softcup application #1, 2. TVUS probe, 3. Lubricant. Further, the vaginal epithelium is clearly hormonally responsive (drastic changes from pre-pubertalreproductive agemenopausal and there are more subtle though still quite perceptible changes in reproductive aged-women through the menstrual cycle as well as with initiation and use/changes in exogenous hormone exposure (hormonal contraceptives)), yet there was no characterization of participants based on age/menopausal status, phase of menses, type of contraception use, etc—all of which are critically important to understand. Thus, the study consisted of an uncharacterized, likely quite heterogeneous in terms of hormonal status, women who all had 3 exposures, and any observed results may have been attributed to any of these exposures, yet the authors conclude that the observed results are based only on the lubricant exposure. This is very problematic.

It would actually be quite interesting to have conducted a study evaluating the epithelial cell numbers and maturity index in cohorts of women separated out by hormonal status: pre-menopausal, naturally cycling in follicular phase; pre-menopausal, naturally cycling in luteal phase; pre-menopausal, on progestogen-only hormonal contraception; pre-menopausal, on combined progestogen/estrogen containing hormonal contraception; post-menopausal. This information could then inform understanding a major factor driving the maturity index, which the authors then used to ‘bin’ the participants. Indeed, the results demonstrate that the only women with significant changes after the ultrasound/lubricant exposure, were those with high MI pre-ultrasound. These women are likely to represent a more hormonally similar grouping, which would be important to know.

The authors state that a strength of the study is the longitudinal within-woman analysis—however the authors do not account for the fact that naturally cycling women and women on cyclic hormonal contraceptives are not in a static state over time with respect to the outcomes of interest—thus, the purported strength is actually a weakness for evaluation of these outcomes. This could have been addressed with a study design that considered hormonal status and timing of collections, however, this study design did not account for these confounding changes.

3. Omission of evaluation of the vaginal microbiota, another significant factor contributing to the vaginal microenvironment. In the discussion, the authors do allude to the microbiota having potential impacts on the MI, which is true—indeed it would be important to understand the contributions to the MI from the hormonal state and microbiological state independently.

4. The statement that a single exposure to vaginal lubricant ‘may increase infection susceptibility’ is overreaching and inappropriate given that the study did not assess this in any way.

Reviewer #2: In this study, the authors used 50 women being exposed to the lubricant EZ, being used for transvaginal ultrasound and took the opportunity to enroll them for pre and several post u/s time points, to see how the lubricant, with a relatively high osmolality, affected cell shedding and maturity. The women self collected cervico vaginal fluid using a Softdisc and froze the samples at home at -20oC until they were brought to the clinic. Clinicians also collected a sample using the same device at 1 visit (Visit 3). Samples were thawed and tested for cell count and maturity index and analyzed by maturity index > or less than 3.

The authors had expected that lubricant exposure would increase shedding of vaginal epithelial (VE) cells. However, at the 6-12 hr time point (1st time post u/s sampled) and several subsequent time points they found less cells, suggesting perhaps they had missed the peak of VE shedding, perhaps within minutes or a few hours, and by the time they sampled what was being shed were deeper layers of less mature cells. Their data do support early increased shedding, that has been observed by others in rectal studies, but they would need to confirm this. Nevertheless, certainly the lubricants are doing something to alter VE balance, finding cells with lower maturity index post u/s in women with higher maturity index (>3) at baseline. The maturity index cut off of 3 was based on an equal prevalence of mature and immature cells in a sample. Having many immature cells (which have low cytoplasm: nuclear ratios) indicates that upper layers (more mature) of VE are lost, potentially exposing deeper layers of VE, reducing protection of the mucosa against abrasion and infections. Hence the interest as to whether the lubricant in this study increased shedding and of which cells.

I found some parts of the manuscript hard to follow and had some questions/clarifications.

Methods

Clarify if exclusion criteria and STI results were self-report, or clinician/lab confirmed.

It is stated that the study continued for 9 weeks, but the paragraph on sample collection and the results only show data for 4 time points (pre, 6-12 h, 1 week, 2 weeks). Was data from post 9 weeks part of the study, and if so clarify where?

What did participants do to carry their -20 oC samples from home to the clinic to keep them frozen?

Cell counts: These were performed on cells thawed from the Softdisc collections. But no DMSO or other cell membrane preservative appears to have been included in the collections/freezing. Could the authors explain why cell counts could be performed using the methods they describe as usually freezing of cells lyses membranes and reduces/alters cell counts? Or perhaps epithelial cells freeze well and can be counted following freezing in absence of DMSO? Similarly for maturity index, is this something that is stable to freeze thaw? If it’s based on cytoplasm/nuclear ratio, if cell membranes rupture during freeze/thaw, is cytoplasm volume altered?

Results.

The text of Fig 2 and 3 legends is a bit confusing. Although the Y axis labels and title of the Fig legend indicate the graphs refer to “change” the text reports “cell count” or “maturity index” not “change in cell count” or “change in maturity index”, so I spent a while looking in the Figs for the actual #s.

Discussion.

The authors could add (end of para 2) that future studies with earlier time points (within a few hours) would confirm the theoretical increased VE shedding that the authors think they missed.

6. PLOS authors have the option to publish the peer review history of their article (what does this mean?). If published, this will include your full peer review and any attached files.

Reviewer #1: No

Reviewer #2: **Yes: **Janet M McNicholl

---

## [Author Response · Author response to Decision Letter 0]

1 Dec 2020

Please see response to reviewer .doc file.

---

## [Decision Letter · Decision Letter 1]

14 Jan 2021

PONE-D-20-18637R1

Observational cohort study of the effect of a single lubricant exposure on cell-shedding from the vaginal epithelium

PLOS ONE

Dear Dr. Brotman,

Thank you for submitting your manuscript to PLOS ONE. After careful consideration, we feel that it has merit but does not fully meet PLOS ONE’s publication criteria as it currently stands. Therefore, we invite you to submit a revised version of the manuscript that addresses the points raised during the review process.

While the revised version has resolved the minor concerns raised by Reviewer #1, our second peer reviewer continues to have substantial concerns regarding the validity of the conclusions drawn. Most importantly, they do not feel that the study setup permits the research team to definitely attribute the genital changes observed to the gel (rather than to the probe). Therefore, they require rewording of the manuscript to reflect this uncertainty, as outlined in their comments below.

We look forward to receiving your revised manuscript.

Kind regards,

Rupert Kaul

Academic Editor

PLOS ONE

Reviewers' comments:

Reviewer's Responses to Questions

**Comments to the Author**

1. If the authors have adequately addressed your comments raised in a previous round of review and you feel that this manuscript is now acceptable for publication, you may indicate that here to bypass the “Comments to the Author” section, enter your conflict of interest statement in the “Confidential to Editor” section, and submit your "Accept" recommendation.

Reviewer #1: (No Response)

Reviewer #2: All comments have been addressed

2. Is the manuscript technically sound, and do the data support the conclusions?

Reviewer #1: No

Reviewer #2: Yes

3. Has the statistical analysis been performed appropriately and rigorously? 

Reviewer #1: Yes

Reviewer #2: Yes

4. Have the authors made all data underlying the findings in their manuscript fully available?

Reviewer #1: Yes

Reviewer #2: Yes

5. Is the manuscript presented in an intelligible fashion and written in standard English?

Reviewer #1: Yes

Reviewer #2: Yes

6. Review Comments to the Author

Reviewer #1: Review for PLOS ONE (PONE-D-20-18637R1)

Title: Observational cohort study of the effect of a single lubricant exposure on cell-shedding from the vaginal epithelium

This study is a non-randomized, single group cohort study of non-pregnant women 18+ years undergoing transvaginal ultrasound for a variety of reasons and willing to collect vaginal fluid using a softcup immediately prior to and at times following the transvaginal ultrasound. The authors evaluated the collected vaginal fluid for number and maturity index of epithelial cells. Overall, the manuscript is greatly improved compared to the original submission.

1. The remaining major concern continues to relate to the study design and attribution of endpoints (epithelial cell number and MI change) solely to the exposure to lubricant, which was not the only exposure and thus attribution cannot be definitively declared as the authors have continued to do. These data were clearly generated from samples collected as a sub-study and as such, the study design is not optimal to definitively state that any observed changes were solely attributable to the gel exposure. This is a common situation for researchers to face and it does not mean that the data are not publishable. It does require clear description and acknowledgement of the multiple exposures, which the current manuscript continues to lack. Even the title of the manuscript directs the reader to believe the conclusion that all of the observed changes are attributable to the single gel exposure, however, there are no controls to demonstrate that the changes did not occur due to the softcup removal of vaginal fluid/cells and/or to the TVUS probe, which were both exposures that all participants in this cohort had. This needs to be clearly acknowledged in the manuscript. I would request a reframing of the title because although there was only a single exposure to gel, the gel was not the only exposure for the vagina. Within the body of the text, the authors did adjust the language of the TVUS probe slightly, to now read “…the effect of a single lubricant exposure incidental to TVUS on cell-shedding from the VE…” This needs to be made even more explicit. I recommend reframing to include the probe in the exposure as follows: “…the effect on cell-shedding from the VE of a single exposure to lubricant gel administered on a TVUS probe … This is truly the exposure. To address the issue with not having a control to assess the impact of the softcup ‘scraping/squeegee’ effect on the vaginal epithelium (for instance, best would be to have had a parallel cohort of women who performed all of the softcup collections at the same times and did not have the gel/TVUS exposure), this needs to be clearly stated in the discussion as a weakness of this study and an acknowledgement that removal of VE cells by softcup squeegee could have contributed to and/or caused the reduction in epithelial cell number and/or MI demonstrated by the data. Indeed, as noted by the authors, this collection method recovers “large volumes of CVF.” The data are what the data are—they simply need to be reported in a non-biased manner with all possibilities for the observed findings considered and discussed.

2. In the abstract, rather than stating, “We determined whether a typical clinical lubricant accelerates shedding of mature epithelial cells, exposing immature cells.”, consider reframing as “We hypothesized that exposure to a typical vaginal lubricant, used for clinical pelvic exams and during transvaginal ultrasonography (TVUS), may contribute to shedding of mature epithelial cells, exposing immature cells. To test this hypothesis, we enrolled women scheduled to undergo TVUS examination for a variety of indications and who were willing to undergo sampling of cervicovaginal fluid immediately before and at three timepoints after the single exposure to lubricant and TVUS (6-12 hours, within one week, and two weeks after).” Please make careful note that the word ‘accelerates’ is problematic as that implies a comparator, and in this study there are none. Same issue in the first sentence of the Discussion, which describes both ‘accelerated’ and ‘increased’ as findings to measures without comparators.

3. Please add references for the statement, “The balance between detachment/shedding and proliferation/maturation processes may influence susceptibility or resistance to reproductive tract infections.”

4. Figure 1. A—Please re-word “as they move up” and “through the parabasal…” as these phrases are non-scientific, vague, and confusing.

5. Figure 1. B—Please do not start with “These” as it is unclear to what this refers. Be specific in your written language.

6. Participants—please add that the exclusion criteria “lubricant use in the week prior to TVUS” was by self-report. Please confirm that at the enrollment visit there was nothing placed in the vagina prior to the softcup collection of vaginal fluid.

7. Was this study registered on ClinicalTrials.gov? If so, please add the registration number to the manuscript.

8. Do you have information about the length of time between the TVUS and the collection of CVF ‘at bedtime on the day of ultrasound’? What was the mean and median number of hours between these two timepoints?

9. Missing 22% of the post-TVUS samples is significant enough to deserve a comment in the weaknesses section of the discussion.

10. Beware that not all ‘hormonal contraception’ is comparable, particularly as it relates to impacting the genital milieu. Women using depot medroxyprogesterone acetate will have markedly thinned epithelia and endometria. Women using hormonal IUDs will also have markedly thinned endometria. Clearly, this study enrolled women using hormonal IUDs. Perhaps adding the various contraceptive methods to Table 1 under ‘Current use of any hormonal contraception’ would be useful.

11. Under ‘Comparison of women with high and low pre-TVUS Maturity Index at W2-post-TVUS’, paragraph starting ‘Maturity index (MI)…’, the verb ‘was’ should be ‘were’ in order to agree with the pleural ‘MIs’.

12. Discussion, paragraph starting, “The epithelial cell changes…” contains a sentence, “These participants began the study with optimal and balanced VE proliferation/maturation and detachment/shedding, the single use of vaginal lubricant incidental to TVUS seems to have caused a significant disruption of that balance.” I don’t see any evidence presented demonstrating “optima and balanced VE proliferation/maturation and detachment/shedding”—what is this based upon? What are the criteria for having an optimal balance? This statement is problematic. Also, the next sentence talks about ‘disruption’, which may or may not be accurate given the absence of data about what is normal at various points in the menstrual cycle. Overall, using language that is purely descriptive, such as “In W1- and W2-post-TVUS samples, we observed persistently low cell numbers and low MIs compared to baseline” and avoiding language that demonstrates bias, such as ‘disruption’, would be preferable. In the discussion it is fair to then hypothesize that the persistently low cell numbers and MI may represent significant and persistent disruption in the epithelium by using language such as this that clearly frames a hypothesis and not stated as fact.

13. Please re-word the vague and grammatically incorrect sentence, “This is not to say that these women’s VE is not affected by lubricant, only that these effects cannot be measured by our methodology.”

14. With respect to the following author responses: “1. Softcup: The softcup was used during all sampling time points and would not impact change in either MI or cell count from the pre-TVUS timepoint to the three post-TVUS time points. In addition, women who experienced a decrease in maturity index or cell count following TVUS had those metrics recover at later timepoints, suggesting that the softcup device is not responsible for the changes observed. Response to Reviewers 2. TVUS probe: It would not be possible to determine the effect of a TVUS probe alone on maturity index or cell count as a TVUS procedure will always include the use of a lubricant in conjunction with the wand. In this sense, the exposure of lubricant + TVUS wand reflects typical use in clinical practice.” 1. Softcup—if the softcup itself (which is an exposure) caused the removal of VE cells, prior to the gel/TVUS probe exposure, the change in MI and/or cell count caused by the softcup would be seen as a change from baseline to follow-up. The removed cells would be in the baseline cup—thus making that measurement ‘high’ relative to the follow-up measurement. 2. Agree that the exposure of the TVUS probe cannot be divorced from the gel, which is precisely why together, they need to be described as the ‘exposure’ and not just the gel.

Reviewer #2: Only request is in the methods section to indicate that the freeze/thaw method is assumed to not affect the assays, based on what the authors said in their response to my question.

Thanks,

7. PLOS authors have the option to publish the peer review history of their article (what does this mean?). If published, this will include your full peer review and any attached files.

Reviewer #1: No

Reviewer #2: **Yes: **Janet McNicholl

---

## [Author Response · Author response to Decision Letter 1]

16 Feb 2021

See response to reviewers in .doc attachments.

---

## [Decision Letter · Decision Letter 2]

8 Mar 2021

PONE-D-20-18637R2

OBSERVATIONAL COHORT STUDY OF THE EFFECT OF A SINGLE LUBRICANT EXPOSURE DURING TRANSVAGINAL ULTRASOUND ON CELL-SHEDDING FROM THE VAGINAL EPITHELIUM

PLOS ONE

Dear Dr. Brotman,

Thank you for submitting your manuscript to PLOS ONE. Based on review of the R1 revised manuscript, we feel that it has merit but does not fully meet PLOS ONE’s publication criteria as it currently stands. Therefore, we invite you to submit a revised version of the manuscript that addresses the minor remaining concerns raised by the peer reviewer.

We look forward to receiving your revised manuscript.

Kind regards,

Rupert Kaul

Academic Editor

PLOS ONE

Journal Requirements:

Reviewers' comments:

Reviewer's Responses to Questions

**Comments to the Author**

1. If the authors have adequately addressed your comments raised in a previous round of review and you feel that this manuscript is now acceptable for publication, you may indicate that here to bypass the “Comments to the Author” section, enter your conflict of interest statement in the “Confidential to Editor” section, and submit your "Accept" recommendation.

Reviewer #1: (No Response)

2. Is the manuscript technically sound, and do the data support the conclusions?

Reviewer #1: Partly

3. Has the statistical analysis been performed appropriately and rigorously? 

Reviewer #1: Yes

4. Have the authors made all data underlying the findings in their manuscript fully available?

Reviewer #1: Yes

5. Is the manuscript presented in an intelligible fashion and written in standard English?

Reviewer #1: Yes

6. Review Comments to the Author

Reviewer #1: Review for PLOS ONE (PONE-D-20-18637R2)

Title: Observational cohort study of the effect of a single lubricant exposure during transvaginal ultrasound on cell-shedding from the vaginal epithelium

This study is a non-randomized, single group cohort study of non-pregnant women 18+ years undergoing transvaginal ultrasound for a variety of reasons and willing to collect vaginal fluid using a softcup immediately prior to and at times following the transvaginal ultrasound. The authors evaluated the collected vaginal fluid for number and maturity index of epithelial cells. Overall, the manuscript is greatly improved compared to the original submission and the R1 submission. The remainder of the comments are minor and intended to further strengthen this manuscript.

Abstract-

1. Please correct the punctuation in the 1st sentence.

2. The first sentence seems disconnected from the rest of the abstract and as currently written the connection of this sentence to the rest of the abstract may not be clear for all readers. This can be resolved by adding the concept that the outer layers are important because…

3. Please re-word the sentence that starts, “The clinical lubricant, EZTM lubricating jelly (Chester laboratory),…” As written, this sentence is awkward. You could say that vaginal lubricants used clinically commonly have high osmolality, pH~5.5, and contain bactericides.

Introduction-

1. Change ‘upwards’ in 3rd sentence—this directionality does not make physiologic sense. Perhaps ‘outwards’ or ‘more distal’ could substitute.

2. 2nd sentence of 2nd paragraph: recommend alternate word choice for ‘seem necessary’

Methods-

1. Participants, 1st sentence: change the tense of ‘seeks’ to match the rest of the sentence that is appropriately past tense.

2. Previous control experiments (RE: freezing of CVF)—please add ‘(data not shown)’

Table 1-contraception-

1. Were all of the oral contraceptive users on estrogen-containing pills? If so, please change to ‘combined oral contraceptives. If not or unknown, keep as oral contraceptives.

Within-woman comparisons-

1. Paragraph 3, 1st sentence: “…indicating depletion of mature…” Please consider changing ‘depletion’ to ‘paucity’ since depletion suggests a change and there is no baseline sampling and thus no comparator. Use of ‘paucity’ also would parallel your appropriate use of ‘predominance’ rather than suggesting that there is an increase.

Comparison of women with high and low pre-TVUS Maturity Index at W2-post-TVUS

1. Please remove the bias from sentence #1. Consider: ‘To assess correlation between the initial epithelial state (high or low pre-TVUS MI) and epithelial state at W2 post TVUS.

2. Maturity Index�it looks like women with low or high pre TVUS MI still had low or high W2 post TVUS MI. Did you assess the change in MI pre�post TVUS? That is, assessing change from baseline for each woman rather than the inter-participant evaluations that you are reporting here? Indeed, this is what is essentially shown in Supplementary Fig 1. Any significant changes? Based on just an eyeball look, it does not appear so, if true, this is important.

Discussion-

1. Second paragraph, sentence 3—please correct grammar. Also, please take great care to not over-state your findings. This study design is imperfect and the findings are not overly convincing, therefore to state as fact, such as “…resulting in depletion of fully mature cells, reduction of cells available for shedding, and increased exposure/loss of immature cells,…” is overstating the findings of this study.

7. PLOS authors have the option to publish the peer review history of their article (what does this mean?). If published, this will include your full peer review and any attached files.

Reviewer #1: No

---

## [Author Response · Author response to Decision Letter 2]

29 Mar 2021

Please see the response to reviewer .doc.

---

## [Editor Report · Decision Letter 3]

1 Apr 2021

OBSERVATIONAL COHORT STUDY OF THE EFFECT OF A SINGLE LUBRICANT EXPOSURE DURING TRANSVAGINAL ULTRASOUND ON CELL-SHEDDING FROM THE VAGINAL EPITHELIUM

PONE-D-20-18637R3

Dear Dr. Brotman,

We’re pleased to inform you that your manuscript has been judged scientifically suitable for publication and will be formally accepted for publication once it meets all outstanding technical requirements.

Kind regards,

Rupert Kaul

Academic Editor

PLOS ONE
---

## [Editor Report · Acceptance letter]

22 Apr 2021

PONE-D-20-18637R3 

OBSERVATIONAL COHORT STUDY OF THE EFFECT OF A SINGLE LUBRICANT EXPOSURE DURING TRANSVAGINAL ULTRASOUND ON CELL-SHEDDING FROM THE VAGINAL EPITHELIUM 

Dear Dr. Brotman:

I'm pleased to inform you that your manuscript has been deemed suitable for publication in PLOS ONE. Congratulations! Your manuscript is now with our production department. 

Kind regards, 

on behalf of

Dr. Rupert Kaul 

Academic Editor

PLOS ONE